# The Effects of SGLT2 Inhibitors on Liver Cirrhosis Patients with Refractory Ascites: A Literature Review

**DOI:** 10.3390/jcm12062253

**Published:** 2023-03-14

**Authors:** Yasunori Miyamoto, Akira Honda, Seiji Yokose, Mariko Nagata, Jiro Miyamoto

**Affiliations:** 1Division of Internal Medicine, Miyamoto Hospital, Inashiki 300-0605, Ibaraki, Japan; 2Division of Gastroenterology and Hepatology, Tokyo Medical University Ibaraki Medical Center, Ami 300-0395, Ibaraki, Japan; 3Dialysis Center, Miyamoto Hospital, Inashiki 300-0605, Ibaraki, Japan

**Keywords:** sodium–glucose cotransporter 2 (SGLT2) inhibitors, decompensated liver cirrhosis, ascites, diuretics

## Abstract

Decompensated liver cirrhosis is often complicated by refractory ascites, and intractable ascites are a predictor of poor prognosis in patients with liver cirrhosis. The treatment of ascites in patients with cirrhosis is based on the use of aldosterone blockers and loop diuretics, and occasionally vasopressin receptor antagonists are also used. Recent reports suggest that sodium–glucose cotransporter 2 (SGLT2) inhibitors may be a new treatment for refractory ascites with a different mechanism with respect to conventional agents. The main mechanisms of ascites reduction with SGLT2 inhibitors appear to be natriuresis and osmotic diuresis. However, other mechanisms, including improvements in glucose metabolism and nutritional status, hepatoprotection by ketone bodies and adiponectin, amelioration of the sympathetic nervous system, and inhibition of the renin–angiotensin–aldosterone system, may also contribute to the reduction of ascites. This literature review describes previously reported cases in which SGLT2 inhibitors were used to effectively treat ascites caused by liver cirrhosis. The discussion of the mechanisms involved is expected to contribute to establishing SGLT2 therapy for ascites in the future.

## 1. Introduction

Ascites is often associated with decompensated liver cirrhosis. Intractable ascites is predictive of a patient’s prognosis [1], and its treatment may improve the outcomes. The treatments for refractory ascites complicated by cirrhosis include aldosterone blockers, loop diuretics [2], and vasopressin receptor antagonists, as well as procedures such as abdominal puncture and cell-free and concentrated ascites reinfusion therapy (CART) [3]. In addition, recent case reports have shown that sodium–glucose cotransporter 2 (SGLT2) inhibitors are effective for refractory ascites.

SGLT2 inhibitors are known to have diabetic and nondiabetic effects. The nondiabetic effects include those on chronic heart failure [4,5,6,7,8,9] and chronic kidney disease [10,11,12,13]. The diuretic effect of SGLT2 inhibitors is considered to be one of their most important nondiabetic actions. The diuretic effects include natriuresis and osmotic diuresis [14], and effects on arginine vasopressin (AVP) [15] and atrial natriuretic peptide (ANP) [16,17] may also contribute to fluid homeostasis. In addition, the impact of SGLT2 inhibitors on the sympathetic nervous system (SNS) and the renin–angiotensin–aldosterone (RAA) system may also be relevant. On the other hand, the improvement of ascites on treatment with SGLT2 inhibitors may be related not only to nondiabetic effects but also to diabetic effects (Figure 1). As antidiabetic agents, they may improve liver function and nutritional status, which may contribute to the reduction of ascites due to increased serum albumin concentrations.

These findings suggest that SGLT2 inhibitors are a useful treatment option in patients with refractory ascites and liver cirrhosis. To date, SGLT2 inhibitors have only been used to treat liver cirrhosis in patients with diabetes mellitus. However, based on recent reports on the effectiveness of SGLT2 inhibitors in the treatment of heart failure and renal failure without diabetes, we believe it is time to discuss the efficacy of these agents in ascites patients without diabetes.

## 2. SGLT2 Inhibitors as Diuretics

SGLT1 and SGLT2 are located in the brush border membrane of the proximal renal tubule and reabsorb glucose filtered by the glomerulus [18]. Approximately 90% of glucose is reabsorbed by SGLT2 located in the early proximal tubule. The remaining 10% of the glucose not absorbed by SGLT2 is reabsorbed by SGLT1 located in the late proximal tubule [19]. SGLT2 inhibitors competitively inhibit the action of SGLT2, causing accelerated urinary excretion of glucose (Figure 2) and a decrease in the blood glucose concentration [19]. In addition, since SGLT2 is a 1:1 cotransporter of glucose and sodium, it has been reported that the administration of SGLT2 inhibitors to patients with type 2 diabetes promotes urinary sodium excretion (natriuresis), resulting in a decrease in the systemic sodium pool [20].

We recently reported a case of refractory ascites due to alcoholic liver cirrhosis treated with an SGLT2 inhibitor [21]. Before the treatment with an SGLT2 inhibitor, this patient was refractory to the conventional diuretics furosemide and spironolactone, and frequent CART was needed. However, after adding an SGLT2 inhibitor to control the patient’s hyperglycemia, the ascites was markedly reduced, and the patient was able to be weaned from CART (Figure 3).

Except for our case, there are two reports in the literature where SGLT2 inhibitors improved refractory ascites or peripheral edema (Table 1) [22,23]. All patients had symptoms associated with liver cirrhosis and diabetes mellitus and were resistant to diuretics. In the case of ascites owing to primary biliary cholangitis (PBC), renal sodium excretion was increased after treatment with an SGLT2 inhibitor. Therefore, natriuresis may have contributed to the improvement of ascites in this patient [22]. In our case, the administration of an SGLT2 inhibitor resulted in approximately 800 mmol of systemic sodium excretion by natriuresis.

Thus, the mechanism of natriuresis by SGLT2 inhibitors differs from that by furosemide and spironolactone, inhibiting sodium reabsorption in Henle’s loop and collecting ducts (Figure 2). Furthermore, urinary glucose is known to cause osmotic diuresis [24]. In an article reporting the efficacy of SGLT2 inhibition (empagliflozin) against acute heart failure, osmotic diuresis is considered to be the primary diuretic mechanism [25]. SGLT2 inhibitors appear to be effective for ascites resistant to furosemide and spironolactone. The ameliorative effect of SGLT2 inhibitors on ascites seems to involve natriuresis and osmotic diuresis, likely involving a mechanism that is different from those of existing drugs.

## 3. Effects of SGLT2 Inhibitors on Serum Sodium Levels

Patients with decompensated liver cirrhosis often exhibit hyponatremia along with refractory ascites, and the severity of the hyponatremia has a significant effect on their prognosis [26,27]. Even mild chronic hyponatremia (serum sodium: 130–135 mEq/L) is associated with adverse outcomes [28]. SGLT2 inhibitors have been suggested to improve hyponatremia [14]. In our decompensated liver cirrhosis case, hyponatremia (serum sodium: 133 mEq/L) was observed before the start of CART until the administration of empagliflozin, an SGLT2 inhibitor [21]. Subsequently, after administering empagliflozin, the serum sodium concentration gradually increased eventually returning to a normal level. Hyponatremia was also normalized after treatment with SGLT2 inhibitors in two other patients with decompensated liver cirrhosis [22,23].

On the other hand, in patients with normonatremia, no further increase in sodium levels was observed with SGLT2 inhibitors [23]. Ohara et al. reported that SGLT2 inhibitors do not increase serum sodium levels in patients with normonatremia [29]. Type 2 diabetes patients treated with canagliflozin showed only modest changes in serum sodium levels [30], and treatment with dapagliflozin did not significantly change the serum sodium levels [31]. In our case [21] mentioned earlier, the excretion of systemic sodium following the treatment with an SGLT2 inhibitor was calculated to be approximately 800 mmol.

Nevertheless, the sodium levels increased and were normalized. Similarly, increased urinary sodium excretion and blood sodium levels were observed in other liver cirrhosis patients treated with SGLT2 inhibitors [21].

Osmotic diuresis induced by SGLT2 inhibitors may explain these strange changes in the serum sodium concentrations. SGLT2 inhibitors promote urinary glucose excretion and are used as antidiabetic agents [32]. Urinary glucose increases renal tubular osmolarity, which keeps urine isotonic by inhibiting sodium and water reabsorption. This is the mechanism underlying SGLT2 inhibitor-induced osmotic diuresis. Since osmotic diuresis involves the excretion of sodium and sodium-independent water [33,34], the serum sodium level does not decrease and instead increases despite sodium excretion.

ANP and AVP may also contribute to the mechanism through which SGLT2 inhibitors do not decrease serum sodium concentrations despite urinary sodium excretion. ANP is a natriuretic peptide secreted from the atrium when the atrial wall is stretched due to increased central venous pressure, such as in heart failure. Administration of empagliflozin, an SGLT2 inhibitor, to a nonhyperglycemic zebrafish heart failure model was shown to decrease the blood ANP levels by lowering central venous pressure [16].

Clinically, the administration of SGLT2 inhibitors to patients with type 2 diabetes was reported to decrease the blood ANP levels [17]. ANP is also a hormone that causes natriuresis, and a decrease in ANP concentration would suppress natriuresis. Therefore, it could be one of the factors preventing excessive sodium excretion and hyponatremia in patients treated with SGLT2 inhibitors.

In addition, the increases in serum sodium concentrations and osmolality caused by SGLT2 inhibitors are thought to stimulate AVP secretion from the posterior pituitary gland and promote water reabsorption in the renal collecting duct [15]. This report showed that AVP secretion under treatment with an SGLT2 inhibitor was positively correlated with the urinary amounts of glucose and sodium. Thus, SGLT2 inhibitors have a regulatory effect that prevents dehydration and hypernatremia due to excessive urine output [15].

Using a bioimpedance analysis device, Ohara et al. evaluated the effects of SGLT2 inhibitors on extracellular water (ECW) and total body water (TBW) in patients with diabetic kidney disease. Unlike loop diuretics and vasopressin V2 receptor antagonists, the rate of ECW reduction after SGLT2 inhibitor treatment was influenced by the pre-treatment ECW/TBW ratio. Thus, SGLT2 inhibitors reduce ECW in patients with a higher ECW/TBW ratio and do not decrease ECW in patients with a smaller ECW/TBW ratio, which indicates that SGLT2 inhibitors are less likely to cause dehydration than other diuretics [29]. It has also been suggested that, compared to other diuretics, SGLT2 inhibitors may cause more water to be excreted from the interstitial fluid than from the intravascular portion of the extracellular fluid [35]. SGLT2 inhibitors reduced ascites more effectively than other diuretics, which may be due to their ability to cause the excretion of more interstitial fluid, as described above.

In conclusion, administering SGLT2 inhibitors to cirrhotic patients with hyponatremia causes both natriuresis and sodium-independent water excretion as a result of osmotic diuresis. Therefore, the serum sodium concentrations do not decrease but instead show an increasing trend. In addition, regulation by ANP and AVP also helps maintain homeostasis of serum sodium levels. However, the reports of SGLT2 inhibitor administration in patients with liver cirrhosis are limited. Detailed studies of the effects of SGLT2 inhibitors on sodium regulation in these patients are awaited.

## 4. Comparison of SGLT2 Inhibitors and Vaptans

Furosemide and spironolactone are the most commonly used diuretics in treating ascites due to cirrhosis. Vaptan is another medicine that can treat ascites refractory to these usual diuretics [36]. Vaptan is an antagonist of the vasopressin V2 receptor in the renal collecting ducts [37] and has a diuretic effect without causing electrolyte excretion (Figure 2). Therefore, although it is effective for ascites with hyponatremia, its use remains controversial because of the risk of hypernatremia. Vaptan is only approved by the European Medicines Agency (EMA) for inappropriate antidiuretic hormone secretion (SIADH) syndrome. On the other hand, although the US Food and Drug Administration (US FDA) has approved its use for liver cirrhosis, it is currently not recommended for use in patients with severe liver damage.

In Japan, tolvaptan is widely used for ascites in patients with liver cirrhosis who are not sufficiently controlled with other diuretics, such as loop diuretics. However, the administration should be initiated under hospitalization because of the potential for rapid dehydration and hypernatremia during induction. Thus, the efficacy of tolvaptan for ascites derived from cirrhosis has been demonstrated in Japan [36,38]. However, no large-scale prospective randomized controlled trials have been carried out, and a meta-analysis of the efficacy of three types of vaptans (tolvaptan, satavaptan, and lixivaptan) did not show an improvement in life expectancy [39].

The treatment with SGLT2 inhibitors is also associated with adverse events, such as urinary tract infections [40]. However, the advantage of SGLT2 inhibitors over vaptan is that they are less likely to cause hypernatremia and have established long-term cardiovascular and renal protective effects [5,6,10]. In considering the effects of SGLT2 inhibitors and vaptan on serum sodium levels, the outcomes of the administration of each drug to SIADH patients (mostly nondiabetic) are helpful. Morris et al. reported that the occurrence of hypernatremia with tolvaptan administration was 25% [41], while only two (2/43: 4.7%) SIADH patients developed hypernatremia when empagliflozin was administered [24]. The risk of hypernatremia with SGLT2 inhibitors is lower than that with tolvaptan. Patients with refractory ascites with cirrhosis are generally older, with multiple comorbidities and medications. Therefore, the long-term cardiovascular and renal protective effects of SGLT2 inhibitors are advantageous compared to vaptan. In addition, SGLT2 inhibitors are safe and well-tolerated in patients with hepatic impairment [42].

## 5. Effects on the Sympathetic Nervous and RAA Systems

The SNS activates receptors in the renal tubules to increase sodium reabsorption. In addition, the SNS activates β receptors in the juxtaglomerular cells to stimulate renin secretion. Reports have shown that plasma norepinephrine concentrations are elevated in patients with decompensated cirrhosis and are inversely correlated with urinary sodium and water excretion [43,44]. Increased plasma norepinephrine concentrations in decompensated cirrhosis are caused by increased secretion rather than decreased hepatic clearance. Therefore, it is suggested that the SNS is activated in patients with decompensated cirrhosis, promoting sodium retention.

There are several reports in the literature regarding the effects of SGLT2 inhibitors on the SNS. The Schlager (BPH/2J) mouse [45,46] is an animal model of neurogenic hypertension. Intraperitoneal administration of the chemical denervation agent 6-hydroxydopamine to the mouse decreases blood pressure. Similarly, when these hypertensive mice were treated with the SGLT2 inhibitor dapagliflozin, the blood pressure decreased, and the norepinephrine levels in renal tissue reduced significantly. The results suggest that SGLT2 inhibitors may inhibit SNS activation [47]. In humans, chronic activation of the SNS has been reported in patients with obesity, metabolic syndrome, and type 2 diabetes [48,49]. Increased muscle sympathetic nerve activity (MSNA) has been observed in obese patients by elevated urinary norepinephrine and its metabolites [48]. Administering regular diuretics to patients with type 2 diabetes increases MSNA with a decrease in circulating blood volume [50]. In contrast, empagliflozin treatment caused a decrease in blood pressure and weight loss due to the diuretic effect, with no change in MSNA or heart rate [51]. As noted earlier, SGLT2 inhibitors appear to act differently from other diuretics concerning sympathetic cardiovascular regulation.

Activation of the RAA system may also contribute to the accumulation of sodium and water in cirrhosis. Wong et al. attributed the initial renal pathophysiological change in sodium retention in cirrhosis to the activation of the RAA system [52]. The activation of the SNS in cirrhosis may promote renin secretion, but increased renal tubular sensitivity to aldosterone has been noted in cirrhosis, which may also contribute to sodium retention [20,53]. In diabetic patients with hypertension receiving angiotensin-converting enzyme inhibitors or angiotensin II receptor blockers, plasma renin activity and aldosterone concentrations were significantly higher after 24 weeks of treatment with dapagliflozin than before treatment [54]. Increases in plasma aldosterone concentrations and renin activity have also been reported when empagliflozin is administered to patients with type 1 diabetes [55] and when dapagliflozin is administered to patients with type 2 diabetes [56]. In contrast, there is a report of no change in renin activity and aldosterone levels in a patient with type 2 diabetes treated with empagliflozin for 5 days [57]. As aforementioned, many reports indicate that SGLT2 inhibitors activate the RAA system. However, it may vary depending on the duration of the administration, and a certain view has yet to be reached.

In addition, all previous reports on the effects of SGLT2 inhibitors on the SNS and RAA system in humans have involved diabetic patients and not patients with liver cirrhosis. Both SNS and RAA systems are expected to be activated in cases of refractory ascites associated with decompensated cirrhosis. The effects of SGLT2 inhibitor use on both pathways will be clarified in the future.

## 6. Effects on Liver Function and Nutritional Status

The 2019 European Society for Clinical Nutrition and Metabolism (ESPEN) and 2019 European Association for the Study of the Liver (EASL) guidelines state that patients with cirrhosis are significantly nutritionally impaired and exhibit protein-energy malnutrition (PEM). The incidence and severity of PEM are associated with liver disease progression and hepatic functional reserve. PEM is diagnosed in around 20% of patients with compensated cirrhosis and in more than 60% of patients with decompensated cirrhosis. In addition, malnutrition is associated with poor prognosis in patients with cirrhosis [58,59].

Our previous report showed that the administration of empagliflozin, an SGLT2 inhibitor, increased the serum albumin levels in a patient with decompensated liver cirrhosis [21]. Subsequently, the serum albumin level normalized, and the serum choline esterase level improved in this patient, suggesting improved liver function and nutritional status (Table 1). In a report using the CONtrolling NUTritional status (CONUT) [60], through the evaluation of the nutritional status calculated from serum albumin, total cholesterol, and blood lymphocyte count, 63.4% of patients with chronic liver diseases were diagnosed as malnourished [61]. The evaluation of our case using the CONUT showed a score of 6 (moderate hyponutrition) before empagliflozin administration, which improved to 2 (mild hyponutrition) after administration. The improved nutritional status may be due to decreased ascites, which improved abdominal distention and increased food intake. Furthermore, the Child–Pugh Score also improved, suggesting an improvement in liver function as well as nutritional status.

The prevalence of diabetes mellitus in patients with cirrhosis has been reported to be 37%, five times greater than in those without cirrhosis [62]. Diabetes mellitus is an independent prognostic factor in patients with cirrhosis. In addition, diabetes is believed to increase the risk of cirrhotic complications such as ascites and hepatic encephalopathy [63]. A previous report [22] showed that the use of an SGLT2 inhibitor improved HbA1c-based hyperglycemia in cirrhotic patients with diabetes. Although there is no direct evidence that the liver function improves when SGLT2 is administered to patients with cirrhosis, SGLT2 inhibitors may have improved the liver function in the examined cirrhotic patients by improving glucose metabolism.

## 7. Effects on Adiponectin

Adiponectin is a 30 kDa protein secreted from adipocytes [64]. Adiponectin suppresses hepatic glycogenesis and decreases tissue triglyceride content by enhancing fatty acid oxidation via activating adenosine monophosphate-activated protein kinase (AMPK) and the peroxisome proliferator-activated receptor α (PPARα). Adiponectin activates 5’-AMP-activated protein kinase, thereby directly regulating glucose metabolism and insulin sensitivity [65].

Garvey et al. showed that canagliflozin increased the serum adiponectin levels by 17% compared to glimepiride in patients with type 2 diabetes [66]. Shiba et al. found that oral administration of canagliflozin to non-alcoholic steatohepatitis (NASH) model mice delayed and inhibited the development of fatty liver, NASH, and NASH liver cancer [67]. They believe that the excretion of glucose from the kidneys into the urine increased the energy storage capacity of the adipose tissue and reduced fat accumulation in the liver. However, adiponectin may also be involved in reducing hepatic fat accumulation.

On the other hand, unlike fatty liver and NASH, the effects of SGLT2 inhibitors on adiponectin in patients with liver cirrhosis and refractory ascites are unknown. However, since anti-inflammatory effects and improvement of apoptosis and liver fibrosis by adiponectin have been suggested [68], SGLT2 inhibitors may have some beneficial effects on decompensated liver cirrhosis through adiponectin.

## 8. Effects on Ketone Bodies

SGLT2 inhibitors increase circulating ketone bodies [69]. It has been reported that dapagliflozin increased plasma β-hydroxybutyrate, a ketone body [70], but no ketoacidosis was observed [31]. Ketone bodies are synthesized from fatty acids, mainly in the hepatic mitochondria under physiological conditions, such as fasting, starvation, ingestion of a ketogenic diet, exercise, and pregnancy. They are transported by the blood circulation to organs throughout the body and used as a source of energy [71]. Ketone bodies also have anti-inflammatory and protective effects on the liver and heart [71]. It has been suggested that increased levels of ketone bodies may contribute to the suppression of cardiovascular death by SGLT2 inhibitors [72].

Acetoacetic acid, another ketone body, inhibits the vascular nucleotide transporter (VNUT) [73]. VNUT is expressed in a variety of cells and has been linked to diabetes [74,75]. VNUT knockout mice show improved glucose tolerance and resistance to type 2 diabetes [76]. In a study of high-fat-diet-induced NASH mouse models, inflammation and fibrosis were markedly reduced in VNUT KO mice [77]. These results suggest that VNUT inhibition by ketone bodies may improve insulin resistance and chronic inflammation in patients with type 2 diabetes.

The ketone body production and anti-inflammatory effects of SGLT2 inhibitors are thought to be pharmacological effects that are distinct from their antidiabetic and diuretic actions. Further detailed mechanistic investigations are awaited.

## 9. Impact of SGLT2 Inhibitors on the Renal Function

SGLT2 inhibitor administration did not significantly alter the renal function, either in our report or in other reported cases with cirrhosis. However, in all patients, ascites and edema decreased within a few months, suggesting that the urine output may have increased in a relatively short period of time. Therefore, it is necessary to ensure that all patients do not have reduced renal function in the early stages of administration.

In EMPA-REG OUTCOME, a study of empagliflozin in type 2 diabetic patients with a history of cardiovascular disease, empagliflozin showed renal protection when observed over time, although a temporary decrease in the renal function was observed [7]. Dapagliflozin was also effective in preventing progression to renal failure, cardiovascular events, and all-cause mortality in patients with and without type 2 diabetes, although the renal function was temporarily impaired [10]. Increased excretion of free water by SGLT2 inhibitors is expected to cause renal dysfunction due to dehydration and should be prevented. In elderly patients, it is necessary to loosen the fluid restrictions to avoid renal damage.

## 10. Conclusions and Future Perspectives

SGLT2 inhibitors were initially developed and marketed for the treatment of diabetes. However, because of their diuretic properties, they are starting to be used for the treatment of chronic heart failure and chronic renal failure. The efficacy of V2 receptor antagonists has been reported in the treatment of refractory ascites inadequately controlled with spironolactone and/or furosemide. However, there are some disadvantages to V2 receptor antagonists, including the risk of liver dysfunction and the fact that they are difficult to use in patients with hypernatremia. On the other hand, SGLT2 inhibitors are well tolerated in patients with impaired liver function and can be used in patients with hypernatremia because they are sodium excretion agents.

Since diabetes is often associated with liver cirrhosis, SGLT2 inhibitors can be used to treat both ascites and glucose metabolism concurrently. It may be necessary to use different diuretics for different conditions, such as SGLT2 inhibitors for ascites in cirrhosis patients with diabetes or hypernatremia. The efficacy of SGLT2 inhibitors in treating refractory ascites due to liver cirrhosis needs to be investigated in more patients in the future. Although only diabetic patients have been included so far, it is desirable to investigate the efficacy of SGLT2 inhibitors for refractory ascites in nondiabetic patients. An open-labeled single-arm trial is currently underway to determine the effect of empagliflozin on ascites (ClinicalTrials.gov Identifier: NCT05013502). As in this ongoing trial, it is necessary to clarify which patient populations, other than diabetic patients, would benefit from SGLT2 inhibitor therapy.

## Figures and Tables

**Figure 1 jcm-12-02253-f001:**
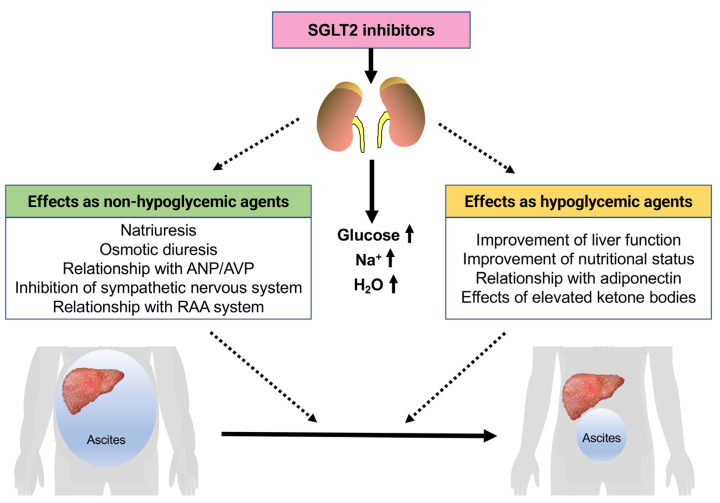
Mechanism of the effects of SGLT2 inhibitors on liver cirrhosis patients with refractory ascites. ANP, atrial natriuretic peptide; AVP, arginine vasopressin.

**Figure 2 jcm-12-02253-f002:**
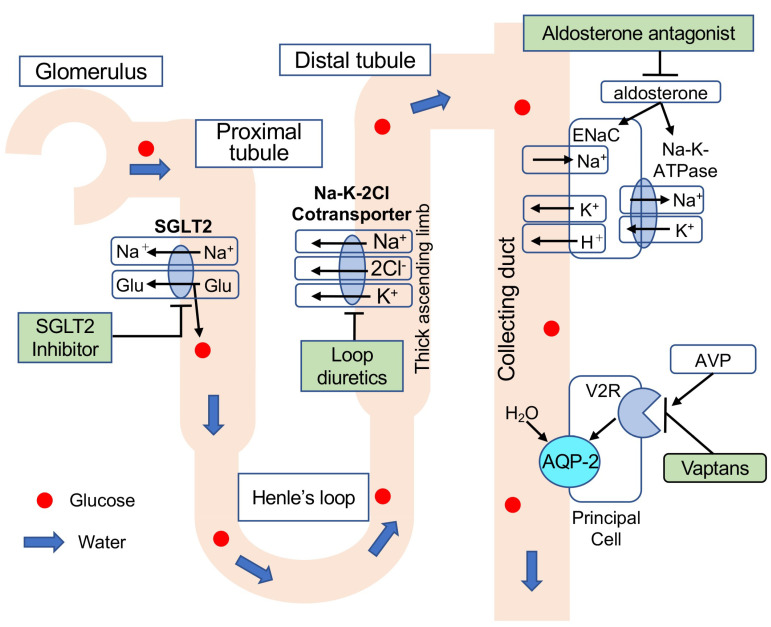
Point of action and mechanism of each diuretic. Loop diuretics inhibit Na^+^/K^+^/2Cl^−^ co-transporters in the ascending limb, thereby inhibiting Na^+^/K^+^ absorption. Diuresis is indicated by increased Na^+^ excretion. Aldosterone binds to aldosterone receptors and subsequently upregulates epithelial sodium channel (ENaC) in the collecting duct, which promotes apical Na^+^ reabsorption. Aldosterone also activates basolateral Na^+^/K^+^-ATPase for Na^+^ excretion from the cell to the interstitial fluid and K^+^ absorption from the interstitial fluid to the cell. Inhibition of these reactions by aldosterone antagonists increases urinary Na^+^ excretion, resulting in diuresis. AVP binds to vasopressin receptor type 2 (V2R) and upregulates the expression and translocation of aquaporin-2 (AQP-2) to the cell membrane on the luminal side of the tubule, resulting in water reabsorption. Vaptans antagonize arginine vasopressin (AVP) and inhibit H_2_O channel activity. Thus, water diuresis occurs.

**Figure 3 jcm-12-02253-f003:**
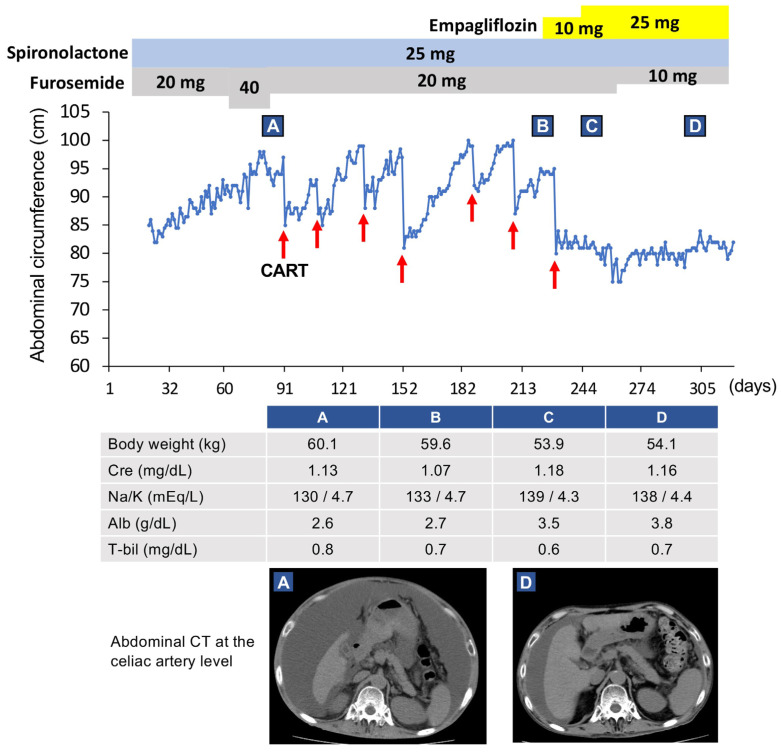
Clinical course of our case. This case was a 59-year-old Japanese man (patient 5 in Table 1). Since his renal function became worse after using diuretics, concentrated ascites reinfusion therapy (CART) was selected. After six rounds of CART, administration of empagliflozin was started. Subsequently, the ascites improved markedly. Red arrows indicate CART administration. Clinical data and radiological examinations (**A**) before CART, (**B**) just prior to empagliflozin administration, (**C**) after using empagliflozin for 28 days, and (**D**) after using empagliflozin for 3 months. This figure was adapted from a previous study [21].

**Table 1 jcm-12-02253-t001:** Characteristics of cirrhosis patients with fluid retention treated with SGLT2 inhibitors.

		Pre	1 Week	1 Month	3 Months	4 Months	6 Months	9 Months
Patient 1 [23] Age: 63 Sex: Female SGLT2-I: Empagliflozin Etiology: NASH	Body weight (kg)	63	58.2	58.2	57.9		58.1	
	Serum Na/K (mEq/L)	139/4.2	140/4.2	137/4.3	136/4.6		135/4.3	
	Serum albumin (g/dL)	2.84	2.86	2.71	2.5		2.88	
	Serum creatinine (mg/dL)	0.7	0.72	0.78	0.78		0.87	
	Platelets (×10^4^/μL)	10.2	10.3	10.4	9.9		9.9	
	Fasting glucose (mg/dL)	86	87	85	73		90	
Patient 2 [23] Age: 64 Sex: Female SGLT2-I: Canagliflozin Etiology: NASH	Body weight (kg)	81.1	80.7	74.8	73.3		69.9	
	Serum Na/K (mEq/L)	120/4.1	140/5.2	138/4.5	145/4.3		141/4.7	
	Serum albumin (g/dL)	3.5	N/A	3.5	3.68		3.38	
	Serum creatinine (mg/dL)	0.7	1.00	0.80	0.81		0.86	
	Platelets (×10^4^/μL)	6.8	N/A	7.0	6.6		7.8	
	Fasting glucose (mg/dL)	140	91	140	141		121	
Patient 3 [23] Age: 53 Sex: Male SGLT2-I: Canagliflozin Etiology: NASH	Body weight (kg)	57.6	55.7	53.5	51.9		51.0	
	Serum Na/K (mEq/L)	135/4.9	139/4.7	139/4.5	139/4.5		145/4.4	
	Serum albumin (g/dL)	3.4	3.1	3.2	3.4		3.6	
	Serum creatinine (mg/dL)	1.80	1.24	1.00	1.04		0.90	
	Platelets (×10^4^/μL)	11.1	16.6	10.5	6.6		9.9	
	Fasting glucose (mg/dL)	187	123	130	119		150	
Patient 4 [22] Age: 54 Sex: Female SGLT2-I: Empagliflozin Etiology: PBC	Body weight (kg)							
	Serum Na/K (mEq/L)	133/4.39	136/4.42	139/4.1		140/3.71		
	Serum albumin (g/dL)	3.1	2.9	3.5		3.7		
	Serum creatinine (mg/dL)	0.84	0.77	0.60		0.55		
	Platelets (×10^4^/μL)							
	Fasting glucose (mg/dL)	286	165	137		116		
	HbA1c (%)	6.6				5.8		
	Child–Pugh score	8	8	6		5		
Patient 5 [21] Age: 59 Sex: Male SGLT2-I: Empagliflozin Etiology: Alcohol	Body weight (kg)	59.6	53.8	54.9	54.1	55.2	55.4	54.7
	Serum Na/K (mEq/L)	133/4.7	136/4.1	139/4.3	138/4.4	137/4.5	141/4.4	141/4.6
	Serum albumin (g/dL)	2.7	3.3	3.5	3.8	3.8	4.0	4.4
	Serum creatinine (mg/dL)	1.07	1.10	1.18	1.16	1.24	1.18	1.16
	Platelets (×10^4^/μL)	8.8	8.2	9.3	10.1	9.8	10.7	12.8
	Fasting glucose (mg/dL)	125	183	144	129	107	137	104
	HbA1c (%)	7.5		7.2	6.4	6.5	6.5	6.1
	ChE (IU/L)	45		72	82	89	93	112
	Child–Pugh score	10	7	7	6	6	6	6

Cirrhosis patients with fluid retention treated with SGLT2 inhibitors. Hyponatremia was normalized after treatment with SGLT2 inhibitors in three patients (patients 2, 4, and 5). No increased sodium levels were observed with SGLT2 inhibitors in normonatremia (patients 1 and 3). In some patients, SGLT2 inhibitor improved liver function and nutritional status (patients 4 and 5). Abbreviations: SGLT2-Ⅰ, sodium–glucose cotransporter 2 inhibitor; NASH, non-alcoholic steatohepatitis; PBC, primary biliary cholangitis; ChE, cholinesterase.

## Data Availability

Not applicable.

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
