# Peer review of "The Effects of SGLT2 Inhibitors on Liver Cirrhosis Patients with Refractory Ascites: A Literature Review"

_jcm, 2023, doi:10.3390/jcm12062253_

Round 1

Reviewer 1 Report

In the manuscript entitled “The Effects of SGLT2 Inhibitors on Liver Cirrhosis Patients with Refractory Ascites: a Literature Review”, Yasunori Miyamoto et al. reviewed the effect of SGLT2 on serum sodium, sympathetic nervous and RAA systems, and liver function, and discussed that SGLT2 inhibitors are a useful treatment option in patients with refractory ascites and liver cirrhosis. The manuscript is well-written, and the topic is very intriguing.

1.   Language needs to be improved throughout the manuscript. For example, in line 44 (Figure 1) and line 73 (Figure.3), please make them consistent.

2.   The authors compare SGLT2 inhibitors and vaptans. There are many drugs to treat cirrhosis complicated by refractory ascites, such as spironolactone and frusemide, authors can compare more drugs with SGLT2, not only vaptans.

3.   The authors mentioned that there are three types of vaptans, but only discussed tolvaptan when compared with SGLT2 inhibitors, please discussed all types of vaptans if the authors can search the related literature.

Author Response

Dear Ms. Nina Qin:

We have revised the manuscript as suggested by the reviewer.

Thank you very much for the careful review of our manuscript.

Sincerely,

Yasunori Miyamoto, M.D., Ph.D.

Replies to the Reviewer #1:

  1. Language needs to be improved throughout the manuscript. For example, in line 44 (Figure 1) and line 73 (Figure.3), please make them consistent.

RESPONSE: We have revised the manuscript as suggested by the reviewer.

  1. The authors compare SGLT2 inhibitors and vaptans. There are many drugs to treat cirrhosis complicated by refractory ascites, such as spironolactone and frusemide, authors can compare more drugs with SGLT2, not only vaptans.

RESPONSE: Thank you for the reviewer’s comment. Furosemide and spironolactone are the most commonly used diuretics in treating ascites due to cirrhosis. Vaptan (especially tolvaptan) is used to treat ascites refractory to these usual diuretics. Since the subject of this review is the treatment of refractory ascites, we focused on comparing vaptan and SGLT2 inhibitors. A comparison of furosemide, spironolactone and SGLT2 inhibitor has been briefly described in Fig. 2.

  1. The authors mentioned that there are three types of vaptans, but only discussed tolvaptan when compared with SGLT2 inhibitors, please discussed all types of vaptans if the authors can search the related literature.

RESPONSE: Thank you for the reviewer’s suggestion. While tolvaptan helps treat refractory ascites, lixivaptan and satavaptan are currently not used for treating ascites because of hepatotoxicity. Therefore, we focused on tolvaptan in this review.

Reviewer 2 Report

Thoughtful article on observations made by the authors in their patients/cases - well written, high quality figures especially Figures 1 and 2, with proposed mechanisms of potential efficacy of SGLT2 inhibitors in decompensated cirrhotic patients with ascites carefully researched in associated related literature.

Author Response

Thank you for the reviewer’s comment.